# Comparative Analysis of Employment Disparities, Precarity and Decent Work between Trans and Cis People in Portugal

José Baptista *, Dália Costa and Sónia P. Gonçalves

Centre for Public Administration and Public Policies, Institute of Social and Political Sciences, Universidade de Lisboa, Rua Almerindo Lessa, 1300-663 Lisbon, Portugal; dcosta@iscsp.ulisboa.pt (D.C.); sonia-goncalves@campus.ul.pt (S.P.G.)
* Correspondence: josediogofbaptista@gmail.com

**Abstract:** While the societal acceptance of trans people has made strides, discrimination remains prevalent in professional settings. The concept of decent work denotes the minimal acceptable standards for the workforce. Conversely, precarity encompasses a multifaceted construct with various indicators, including unemployment. Achieving decent work necessitates the absence of discrimination, ensuring inclusivity for all individuals. Nevertheless, when trans individuals encounter discrimination in professional contexts, and considering the interconnectedness of precarity and gender identity, along with the literature suggesting elevated levels of unemployment among trans people, it becomes crucial to explore their professional integration experiences. Consequently, this study seeks to compare disparities in unemployment, precarity and decent work between trans and cis individuals. A questionnaire, featuring previously validated instruments (Decent Work Scale, $\alpha = 0.86$, and Employment Precariousness Scale II, $\alpha = 0.86$), along with custom questions, was administered to a sample of 202 participants (97 trans and 105 cis) between 11 October 2022 and 14 November 2022. The collected data underwent analysis using SPSS (Statistical Package for the Social Sciences) Statistics 28 and Mplus. The results underscored distinctions between each group's professional realities, concluding that trans people exhibit higher rates of unemployment and precarity while experiencing lower levels of decent work when compared to cis people. This prompts inquiries into the factors contributing to these differences and an exploration of the consequences of trans individuals limited professional integration.

**Keywords:** decent work; human resources; organizational diversity; precarity; trans; unemployment

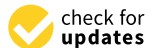



## 1. Introduction

Gender is an essential dimension of human identity, expressed as an individual reflection—masculine, feminine, both or none (Cobb and McKenzie-Harris 2019). The world today is considered cisnormative, meaning there is an assumption that the entire population is cis. However, in a global context, it is estimated that between 0.4% and 1.3% of the population that is over 15 years of age does not identify with the gender assigned at birth, representing around 25 million people worldwide (Winter et al. 2016).

The term trans is a broad term, referring to all people whose identity and/or gender expression differs from the gender categories assigned at birth (Winter et al. 2016). This term describes a diverse group of people, including people who wish to be part of the binary system and others who do not have that intention, as non-binary individuals (Kattari et al. 2022). In opposition, the term cis refers to those who have a gender experience that is congruent with the gender and sex assigned at birth (Bauerband et al. 2018).

The relationship between gender and professional opportunities is intricate, as there is evidence that suggests that an individual's job performance is assessed through the lens of their gender (Schilt 2006). The limited presence of trans individuals in the labor market is a cause for concern, as it hampers the development of a united front capable

of driving change and promoting better integration of gender diversity. This, in turn, would reduce precarity and instances of discrimination in professional environments (Barclay and Scott 2006). Assuming that individuals possess equivalent human capital, it becomes crucial to explore the existence of mechanisms that perpetuate gender stereotypes and their repercussions in the workplace. Such mechanisms result in organizations not integrating trans people in the same manner as cis people (Rudman and Glick 2008).

It is acknowledged that the persistent binary perception of gender within various facets of Western societies results in a significant portion of the population remaining uninformed about the existence of trans people. Consequently, this lack of awareness often translates into limited or no interactions with trans people for many individuals. In the Portuguese context, trans people are perceived by the general population as the most discriminated group in the country, encountering significant barriers in accessing employment (Baptista et al. 2023b; Costa et al. 2010; Saleiro 2013). The objective of this study is to understand the integration experiences of trans people in a professional context, analyzing the differences between the levels of unemployment, precarity and decent work amongst trans and cis people in Portugal.

Work is a central element in everyone's life, as a source of income and survival, or due to its regulating effect on social life, impacting dimensions such as: survival/basic needs; identity; family; social inclusion; health; well-being; and quality of life (Ferreira et al. 2019; Ramalho and Costa 2017). The concept of decent work emerged in 1919, considering aspirations for professional life and defining work as something that is productive and where workers rights are protected, providing a salary and adequate social protection (Ferraro et al. 2021; Knox 1919). According to the International Labor Organization (ILO), decent work consists on the minimum acceptable standards for the working population, including the following elements: a safe working environment (absence of physical, psychological or emotional abuse); access to adequate health care; adequate remuneration; suitable working hours that allow time for rest and free time; and organizational values congruent with social and family values (Duffy et al. 2017; International Labor Organization (ILO) 2008, 2019).

Precarity and insecurity felt in the labor market can have a short- or long-term impact on the health of the workforce (Caldbick et al. 2014). The labor market functions under a law of demand and supply, in which the population that works or is searching for work possess two competences: employed people and unemployed people (Hussmanns 2007). Unemployment (as a component of precarity) can be defined as a situation associated with the total absence of work or absence of work that is regulated through a contractual relationship that institutionalizes work into employment (Rodrigues et al. 2017). In opposition to decent work, precarious work refers to temporary, insecure or part-time work that is generally poorly paid, without benefits or minimal legal and social protections. This type of work has undergone a significant increase during the various economic crises, affecting mostly minority groups (Blustein et al. 2020).

Trans people report difficulties in finding and/or maintaining a job due to discrimination related to their gender identity and/or expression, registering unemployment levels approximately twice as high when compared to cis people (Grant et al. 2011). Thus, there is an under-representation of trans people in organizations, with not much official data. In the literature, trans people are more often studied from a perspective of instability, invisibility, vulnerability and violence, and very few times addressed from a perspective of access to rights, namely in terms of access to employment (Almeida and Vasconcellos 2018).

Upon entering the labor market, trans individuals pursue careers in diverse sectors and fields, confronting a multitude of personal, professional and legal obstacles. Despite the increase in the public presence of trans people in entertainment environments and in media in general, and as such a consequent increase of awareness of the barriers faced by this population, finding a job is challenging for trans people, and if they achieve this goal, they are more at risk to experience different forms of discrimination, namely transphobia, inappropriate jokes and language or harassment, making it difficult to remain employed under these conditions and making it difficult for organizations to maintain a healthy

work environment when there is an unequivocal violation of rights. Adding to all this, the non-existence or lack of adequacy of organizational policies and/or laws that protect trans people makes the process of integration in the labor market even more complex (Baptista et al. 2023a; Barclay and Scott 2006; Nadal et al. 2014). Transphobia is the term used to describe discrimination and bias that is specifically linked to gender identity, including unease, fear, hate, repulsion and prejudiced treatment against all people who express identities and/or have gender expressions that are not normative (Hill 2002; Hill and Willoughby 2005).

The present study intends to compare the levels of decent work, precarity and unemployment between trans and cis individuals in Portugal. In order for decent work to exist, the absence of discrimination and consequent inclusion of all people is necessary (Bletsas and Charlesworth 2013). However, if trans people experience high levels of discrimination in professional contexts (Cobb and McKenzie-Harris 2019); if the notion of precarity is intricately linked to the concepts of gender identity (Puig-Barrachina et al. 2014); and if the literature suggests that trans people are more likely than cis people to be unemployed (Conron et al. 2012), two hypotheses arise that relate all these variables:

**H1.** *Trans people have lower levels of decent work compared to cis people;*

**H2.** *Trans people have higher levels of precarity and unemployment compared to cis people.*

## 2. Materials and Methods

In order to achieve the objective of this study, previously validated instruments were applied (quantitative method with a deductive approach). To analyze the levels of decent work and precarity of the samples, two different scales were applied (in a questionnaire format): the Decent Work Scale—DWS, $\alpha = 0.86$ (Duffy et al. 2017) and the Employment Precariousness Scale II–EPRES-II, $\alpha = 0.86$ (Vives et al. 2010). As each person will perceive different levels of decent work and precariousness, these two constructs are inherent to the perceived working conditions (Blustein et al. 2020).

The DWS was applied in its full and unaltered format, using the validated Portuguese version, $\alpha = 0.81$ (Ferreira et al. 2019). This instrument consists of 15 items based on the definition of decent work and its corresponding five elements from the International Labour Organization (ILO), which are represented as subscales (composed of three items each): (1) safe working environment (absence of physical, psychological or emotional abuse); (2) access to adequate healthcare; (3) adequate remuneration; (4) suitable working hours that allow rest and leisure; and (5) organizational values congruent and aligned with social and family values (Duffy et al. 2017; International Labor Organization (ILO) 2019). The response format of this scale consists of a seven-point Likert scale, where high scores reflect higher levels of decent work and low scores reflect lower levels of decent work (Duffy et al. 2017; Likert 1932).

The EPRES-II is a revised version of the Employment Precariousness Scale (EPRES), translated from Spanish to Portuguese following the cross-cultural translation procedures and best practices (Brislin 1986; Amable 2006; Vives et al. 2015). EPRES-II consists of 22 questions and 6 dimensions, each representing a subscale: (1) temporariness—contractual relationship and duration; (2) disempowerment—level of negotiation of working conditions; (3) vulnerability (to authoritarian treatment); (4) salary—low or insufficient, with the possibility of economic deficiencies; (5) rights—eligibility for labor rights and social security benefits; and (6) exercise of rights—practical powerlessness to exercise work-related rights (Vives et al. 2010, 2015). The response method for this instrument is based on a three or five-point scale (depending on each item), with total item scores ranging from zero (not precarious) to four (very precarious). This instrument was not applied in its entirety, and only the dimensions suitable for the specific objective of this study were selected—in this case, the salary dimension and the vulnerability dimension (Vives et al. 2010, 2015).

Self-composed questions were also added, and socio-demographic data were also collected in order to analyze other constructs (e.g., unemployment). After the questionnaire

was finished, a pre-test was conducted online, on 10 October 2022. Later, the questionnaire was shared, reaching out for voluntary responses. The sample method was non-probabilistic, as it is not possible to guarantee that all the people have the same possibility of being contemplated in the sample, not allowing for a generalization of the results for all trans and cis people in Portugal. The questionnaire consisted of 34 questions, 5 of a sociodemographic nature. Data collection began on 11 October 2022 and ended on 14 November 2022.

The sample consists of 97 trans people (32 trans women, 51 trans men and 14 trans non-binary people), with an average age of 28.91 years (SD = 8.179); and 105 cis people (68 cis women and 37 cis men), with an average age of 30.68 years (SD = 7.217). The inclusion criteria were that the person's age was 18 or above, that they work and reside in Portugal and identify as trans or cis.

Data collection ended when a satisfactory number of responses was reached in order to consider the sample significant, corresponding to a minimum value of five responses per item (Hair et al. 2013). According to the complete instrument applied with the largest number of items (DWS—with 15 items), the ideal number of participants would be 75 people, and this number was exceeded. The collected data were analyzed using the SPSS (Statistical Package for the Social Sciences) Statistics 28 software (IBM Corp.) and Mplus (Muthén 2021).

The sample was accessed through online channels, including messaging, emails and other sharing methods. The research procedure adhered to ethical standards, ensuring the confidentiality and anonymity of each participant in compliance with the General Data Protection Regulation 2016/679 of the European Union, which has been in effect in Portugal since 25 May 2018. Participation in the study was entirely voluntary and unpaid. Moreover, the research received ethical approval from the Ethics Committee of ISCSP—Higher Institute of Social and Political Sciences of the University of Lisbon in January 2022, documented as deliberation CE-01-2022.

In order to analyze the collected data, a content analysis of the responses was carried out. With regard to the only instrument used in its entirety (DWS), an analysis of the psychometric characteristics (Cronbach's alpha) was carried out, as well as the respective construct validity tested (Confirmatory Factor Analysis—CFA). Subsequently, differences between trans and cis people answers were analyzed using t-student tests.

## 3. Results

Proceeding with the CFA of the scale, the bifactorial solution of DWS, proposed by Duffy et al. (2017) and corroborated by Ferreira et al. (2019) for the respective Portuguese validation, was initially verified. The results confirmed a satisfactory factorial solution ($\chi2$ = 88,862; RMSEA = 0.03; CFI = 0.99; TLI = 0.99; WRMR = 0.49), confirming the construct validity of the scale and subscales.

Continuing with the content analysis of the data collected, with regard to discrimination associated with gender (identity and/or expression), when asked how often trans people experience discrimination for reasons associated with their gender identity/expression, it was possible to perceive that only 12.4% (*n* = 12) have never felt this type of discrimination, with 7.2% (*n* = 2) always feeling it, 19.6% (*n* = 19) feeling it often, 37.1% (*n* = 36) sometimes and 23.7% (*n* =23) rarely. It was also possible to perceive that trans women are the ones who report a higher percentage regarding discrimination as a constant, with 12.5% (*n* = 4) selecting the option always, followed by trans non-binary people (7.1%, *n* = 1) and trans men (3.9%, *n* = 2). Comparatively, 45.7% (*n* = 48) of cis people never felt this discrimination and no one who identifies as cis in the sample reported feeling it always, where only 4.8% (*n* = 5) feel it often, 21% (*n* = 22) sometimes and 28.6% (*n* = 30) rarely. Regarding the contexts where discrimination is experienced, 29.9% (*n* = 29) of trans people reported experiencing this type of discrimination in a professional context, compared to 22.9% (*n* = 24) of cis people.

Regarding the professional status of the trans individuals, 16 people (14%) are unemployed—1.8% (*n* = 2) looking for their first job and 12.2% (*n* = 14) looking for a new job. With regard to the long-term unemployment for a period equal to or greater than two years, 22.7% (*n* = 22) have already been in this situation and 56.7% (*n* = 55) have never been in this situation. Comparing, cis people reported only 4.2% (*n* = 5) of unemployment (looking for a new job—none in the situation of looking for a first job). Regarding the long-term unemployment for a period equal to or greater than two years, it is possible to see that 10.5% (*n* = 11) of cis people have already been in this situation and that 84.8% (*n* = 89) have never been in this situation.

In what concerns the levels of job insecurity, analyzed using two of EPRES-II subscales (salary and vulnerability), where high levels in the respective rating will represent high levels of job insecurity, a content analysis was carried out for each question, since the instrument was not applied in its entirety. Analyzing the specific questions of the salary subscale, it was possible to perceive that the average monthly salary (in net value) for trans individuals (excluding the 12 people who were not comfortable answering that question) is in the range between EUR 601 and EUR 750 (net value) per month, with the most representative salary range being between EUR 601 and EUR 750 (net value) per month. It is also possible to point out that 14.4% of trans people (*n* = 14) live with a net salary equal to or less than EUR 300 per month and that only 1% (*n* = 1) have a net salary greater than EUR 3000 (net value) per month. In turn, for cis individuals (excluding the two people who were not comfortable answering that question), the average net salary per month is in the range between EUR 1201 and EUR 1500, and the salary range with the highest representation is between EUR 751 and EUR 999 euros (net value) per month. It is also possible to highlight that only 1.9% of the cis people (*n* = 2) live with a net salary equal to or less than EUR 300 per month and that 6.7% (*n* = 7) have a net salary greater than EUR 3000 per month.

Finally, analyzing the EPRES-II vulnerability subscale, it was possible to obtain the following results: 27.8% (*n* = 27) of trans people are always afraid to demand better working conditions, compared to only 9.5% (*n* = 10) of cis people; and 16.5% (*n* = 16) of trans people always feel defenseless in the face of unfair treatment by people in higher positions, compared to only 2.9% (*n* = 3) of cis people. In both trans people (52.6%, *n* = 51) and cis people (65.7%, *n* = 69), most answered that they would never be fired if they did not do what was asked of them; 39.2% (*n* = 38) of trans people reported never being treated authoritatively, compared to 45.7% (*n* = 48) of cis people; and 27.8% (*n* = 27) of trans people always feel as if they could be easily replaced, compared to only 6.7% (*n* = 7) of cis people.

## 4. Discussion

The results emphasize the employment disparities between trans and cis people, leading to the conclusion that trans people show higher levels of unemployment and precarity and lower levels of decent work, confirming the following hypotheses—H1: trans people have lower levels of decent work compared to cis people; and H2: trans people have higher levels of precarity and unemployment compared to cis people.

In what concerns the unemployment levels of the sample, trans people showed 14% of unemployment (*n* = 16), compared to 4.2% in cis people (*n* = 5), highlighting the employment disparities between trans people and cis people. As being trans is an individual characteristic that is not related to the capacity to do any type of work, it would be expected that this characteristic should not affect the professional realities of trans people, which is not the case (McFadden and Crowley-Henry 2016). Trans people report difficulties finding and maintaining a job due to the discrimination surrounding their identity and/or gender expression, registering unemployment levels almost twice as high when compared with cis people (Grant et al. 2011), in this case more than double (actually more than triple).

Trans people are more at risk to feel that the professional context is not a safe and welcoming place, as it is a space where they are more at risk to experience diverse discriminatory attitudes and behaviors (Davis 2009; Whittle and Turner 2017). The literature

suggests that it is in the professional context where (on a European level) the highest levels of prejudice and discrimination against trans people are reported, not only when searching for a job but also in the workplace, along with the levels of unemployment (Beauregard et al. 2021). When looking at the results concerning experienced prejudice and discrimination associated with gender (identity and/or expression), when questioned about the frequency with which they experienced this type of discrimination, it was possible to understand that only 12.4% (*n* = 12) of trans people have never felt this discrimination, compared to 45.7% (*n* = 48) of the cis people. In addition, 7.2% (*n* = 2) of trans people always feel this type of discrimination, compared to no cis people in the sample.

Regarding decent work, the results are once again aligned with what the literature suggests, when it states that the absence of decent work may be associated with poverty, a lack of job security and gender-related discrimination (Blustein et al. 2020). Regarding the levels of decent work, accessed through the DWS scale (Duffy et al. 2017), it was possible to perceive that, on average, cis people have higher levels of decent work (M = 66.27, SD = 16.331) compared to trans people (M = 55.82, SD = 18.483), and it was also possible to confirm, through a t-test, the existence of statistically significant differences between groups—proving that the results of this scale can be justified and attributed to the construct of gender identity.

When analyzing the levels of professional precarity through two EPRES-II subscales (salary and vulnerability), it was possible to perceive that, the average salary per month (in net value) of the trans people in the sample (excluding the 12 people who did not feel comfortable answering this question) is in the range between EUR 601 and EUR 750, compared to the average salary per month (in net value) of the cis people in the sample (excluding the two people who did not feel comfortable answering this question), which is in the range between EUR 1201 and EUR 1500. It is necessary to consider the Portuguese minimum wage, legally know as Minimum Monthly Retribution Guaranteed, which represents the minimum amount of monthly salary in Portugal (excluding food, holiday or Christmas allowances). On the date of January 2023, according to Decree-Law No. 85-A/2022, of December 22 (published in Diário da República), this value corresponds to EUR 760.00 gross (EUR 676.40 in net value). In the results obtained, it was possible to perceive that the average salary per month (in net value) of the trans people in the sample is in the range between (net) EUR 601 and EUR 750 (and may, therefore, be lower than the value of the Portuguese minimum wage in net terms). Comparatively, the average monthly net salary of the cis people in the sample is in the range between EUR 1201 and EUR 1500 in net value (well above the Portuguese minimum wage in net terms).

Lastly, in the vulnerability subscale, it was possible to see higher values for trans people compared to cis people (e.g., 27.8% (*n* = 27) of trans people are always afraid to demand better working conditions, compared to only 9.5% (*n* = 10) of cis people; or 16.5% (*n* = 16) of trans people always feel defenseless in the face of unfair treatment by people in higher positions, compared to only 2.9% (*n* = 3) of cis people). In this way, it is possible to perceive that the trans people in the sample have higher levels of precarity, compared to the cis people in the sample.

## 5. Conclusions

In summary, as we delve into the overarching objectives of this study, which aimed to conduct a comparative analysis of unemployment, precarity and levels of decent work between trans and cis individuals in the Portuguese context, the findings serve to underscore the significant disparities in employment experiences between these two distinct professional realities. This, in turn, leads to a compelling conclusion: trans individuals are more at risk of exhibiting elevated rates of both unemployment and precarity, alongside diminished levels of decent work. These outcomes not only prompt inquiries into the underlying factors contributing to such disparities but also encourage the consequences of the integration of trans people into the workforce to be explored, specifically in terms of their unemployment rates, experiences of precarity and access to decent work. In light of these

observations, it becomes apparent that the initial hypotheses have been confirmed—H1: trans people have lower levels of decent work compared to cis people; and H2: trans people have higher levels of precarity and unemployment compared to cis people.

When looking at the limitations of this study, they can be used as leads for future investigation. The main limitation is related to the small dimension of the sample. According to Waite (2020), this represents the biggest impediment when investigating work and employment experiences of trans people, hence the reduced literature in this area. In this sense, the reduced dimension of the sample represents a limitation when looking to generalize results, so it would be interesting in future research to replicate the study with a larger sample, considering a broader set of experiences and realities. Additionally, the use of a nonrandom sample also represents a limitation, as well as the use of trans people in the sample without considering or focusing on the specificities of diversity or distinguishing the results of each variable for trans women, trans men or other diverse identities. It is complex to talk about the identities and gender expressions of trans people without distinguishing them in their diversity, as it is similar, in conceptual and analytical terms, to referring to the cis population without distinguishing between cis women and cis men (Saleiro 2013).

In future research endeavors, it could prove both intriguing and beneficial to replicate the utilization of the used instruments on a more expansive and randomly chosen sample. Doing so would augment the scope of the findings, affording a broader perspective for generalizing the results. Moreover, there is an opportunity to explore various avenues of inquiry by conducting separate studies, each with a specific focus. This could entail investigations centered exclusively on trans women, trans men, trans non-binary individuals or those belonging to other diverse trans identities. This diversified approach could yield valuable insights into the nuanced experiences of these distinct groups. Furthermore, expanding the study's geographical scope by replicating it in different countries holds potential for illuminating variations in trans experiences and shedding light on the influence of cultural factors. By undertaking cross-cultural comparisons, it would be possible to gain a deeper understanding of how cultural contexts shape the dynamics of the integration of trans people into the workforce. This multifaceted approach to future research promises to enhance the comprehension of this multifaceted topic and contribute to a more comprehensive body of knowledge.

**Author Contributions:** Conceptualization, J.B., D.C. and S.P.G.; methodology, J.B., D.C. and S.P.G.; software, J.B.; validation, D.C. and S.P.G.; formal analysis, J.B., D.C. and S.P.G.; investigation, J.B., D.C. and S.P.G.; resources, J.B., D.C. and S.P.G.; data curation, J.B.; writing—original draft preparation, J.B.; writing—review and editing, J.B., D.C. and S.P.G.; visualization, J.B., D.C. and S.P.G.; supervision, D.C. and S.P.G.; project administration, J.B., D.C. and S.P.G. All authors have read and agreed to the published version of the manuscript.

**Funding:** This research received no external funding.

**Institutional Review Board Statement:** The study was conducted in accordance with the Declaration of Helsinki and approved by the Ethics Committee of ISCSP—Higher Institute of Social and Political Sciences of the University of Lisbon (protocol code CE-01-2022, January 2022).

**Informed Consent Statement:** Informed consent was obtained from all subjects involved in the study.

**Data Availability Statement:** The data presented in this study are available on request from the corresponding author.

**Conflicts of Interest:** The authors declare no conflict of interest.

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
