# Peer review of "Comparative Analysis of Employment Disparities, Precarity and Decent Work between Trans and Cis People in Portugal"

_socsci, doi:10.3390/socsci12090510_

Round 1

Reviewer 1 Report

Excellent. I should improve the text explaining only the concept of patriarchy´s biological foundations appear to be so very insecure, and the socialization and link with some neofascist governments in Europe and USA (See  Kate Millett, Sexual Politics, 2016) society and the hate to transpeople as well as the negative reaction of some feminist movement againts men trans process. All the methodology is perfectly explained and developed. Results and conclusions are really interesting and innovative for this field of research. 

Reviewer 2 Report

Thank you very much for allowing me to review this very interesting article. Here are some suggestions for improvement according to sections.

In the title it would be interesting to add the word comparison because I understand that it summarises better the objective of the study which has been comparative.

In the summary it would be interesting to add a methodology that explains how these data have been measured and what scales have been used.

The part of justification, objectives and hypotheses is better at the end of the introduction than in the methodology. In the introduction I also find it interesting to talk about the theories in relation to inclusive behaviours because this is what shows that the problem of discrimination is related to this problem. From the way the article is presented, I consider that the aim is more to compare than to analyse.

In the methodology, it would be better to explain the instruments and why a translated and non-validated scale has been used and why certain items have been chosen. It is not clear. Improve this section by ordering for example following the scheme: design, participants, instruments, procedure and data analysis.

In the results there is part of the methodology section, explain if the objective is also to validate a scale by the cfa.

In the discussion start by explaining whether the objectives and hypothesis have been achieved and not theory. Part of the conclusions corresponds to this section, as it compares with other theories and the last paragraph is the conclusion.

Put this part before the limitations and future studies and point out the strengths for what they are.

Revise the citation because there are errors and there seem to be two citation formats and no one more up-to-date citations.

Round 2

Reviewer 2 Report

Thank you very much for reviewing the article and responding to the improvements indicated in the review.

Author Response

Thank you for your review.

With kind regards,